# Isolation and Structural Identification of New Diol Esters of Okadaic Acid and Dinophysistoxin-1 from the Cultured *Prorocentrum lima*

**DOI:** 10.3390/toxins17010028

**Published:** 2025-01-07

**Authors:** Yeong Kwang Ji, Semin Moon, Sangbum Lee, Yun Na Kim, Eun Ju Jeong, Jung-Rae Rho

**Affiliations:** 1Department of Oceanography, Kunsan National University, 558 Daehak-ro, Gunsan 54150, Republic of Korea; kwang7089@kunsan.ac.kr (Y.K.J.); xennin@kunsan.ac.kr (S.M.); sblee08@kunsan.ac.kr (S.L.); 2Department of Plant & Biomaterials Science, Gyeongsang National University, Jinju 52725, Republic of Korea; skdbssk@hanmail.net

**Keywords:** *Prorocentrum lima*, diol ester, cytotoxicity evaluation, diol diester derivative

## Abstract

*Prorocentrum*, a dinoflagellate responsible for producing diarrhetic shellfish poisoning (DSP) toxins, poses significant threats to marine ecosystems, aquaculture industries, and human health. DSP toxins, including okadaic acid (OA), dinophysis toxin (DTX), and their diverse derivatives, continue to be identified and characterized. In this study, we report the isolation of four new diol esters of OA/DTX-1 from large-scale cultures of *Prorocentrum lima*. Their chemical structures were elucidated through comprehensive NMR and MS analyses, along with structural comparisons with the well-known OA. Notably, compound **1** featured an additional ester group within the diol unit, while compound **2** was revealed to be a C11 diol ester. The cytotoxicity of these newly isolated derivatives was evaluated against three cell lines: Neuro2a (mouse), HCT116 (human), and HepG2 (human). All diol esters exhibited cytotoxic effects, with compound **3** displaying toxicity comparable to OA. These results expand our understanding of DSP toxin diversity and provide valuable insight into the structural variations and biological activity of diol esters of OA/DTX-1.

## 1. Introduction

Dinoflagellates of the genus *Prorocentrum* are well-known producers of diarrhetic shellfish poisoning (DSP) toxins, including okadaic acid (OA) and dinophysistoxins (DTXs), similar to *Dinophysis* species [1,2,3]. These toxins, particularly OA and DTX-1 and -2, are potent inhibitors of protein phosphates PP1 and PPA2 [4]. Consumption of shellfish contaminated with these toxins caused gastrointestinal issues such as diarrhea, nausea, and vomiting, as well as more significant health problems [5]. Over the past few decades, extensive research has focused on DSP toxins and their derivatives produced by these harmful marine algae.

Studies have identified a wide variety of derivatives of OA and DTXs in cultured *Prorocentrum* species, expanding our understanding of their chemical diversity [6,7,8,9,10,11]. Representative derivatives include water-soluble sulfated diesters and diol esters with carbon chains ranging from C6 to C10. Sulfated diesters have been isolated using rapid extraction techniques or cellular boiling, leading to the identification of nine sulfated diesters (DTX-4a, 4b, 5a, 5b, and 5c from OA and DTX-7a, 7b, 7c, and 7d from DTX-1) [12,13,14,15]. In contrast, diol esters have been isolated through mild solvent extraction of cellular biomass (Appendix A) [6,8,9,10,11,15,16,17,18,19,20], with new variants continually being discovered [16].

These derivatives are believed to serve as a self-protection mechanism for toxin-producing organisms [12,15]. Sulfated diesters are hypothesized to act as precursors, hydrolyzing into diol esters and subsequently converting into the toxic free forms of OA and DTXs when cells are ruptured or damaged. Recent research supported this proposed mechanism [15]. In the context of OA and DTX transformations, the toxicity of intermediate diol esters can be posed as a question. While some studies have demonstrated the toxicity of diol esters in experimental systems, such as the diatom *Thalassiosira weissflogii* and in mouse models [17,21], data on their toxicity remain limited.

Interestingly, diol esters are often detected in higher quantities than free OA and DTXs [21,22]. Their composition appears to vary among across *Prorocentrum* strains from different geographic regions [23,24], suggesting ecological significance and potential physiological functions. This variability offers an opportunity to estimate strain-specific differences in *Prorocentrum* species by analyzing toxin profiles that include free OA, DTXs, and diverse diol esters.

During our investigation into toxin profile, we isolate three new OA diols and one DTX-1 diol ester from large-scale laboratory cultures of *Prorocentrum lima*. Here, the chemical structures of these compounds will be elucidated using NMR and MS techniques to reveal a remarkable structural diversity among diol esters. Furthermore, cytotoxicity tests across three cell lines will be conducted to assess the relative toxicity of the isolated diol esters compared to OA.

## 2. Results

The *P. lima* culture was harvested and extracted with MeOH. The extracts were then subjected to a series of chromatographic separations, resulting in the isolation of four new diol derivatives of OA/DTX-1, as shown in Figure 1.

### 2.1. Structure Determination of Compounds ***1***–***4***

The molecular formula of **1** was isolated as a colorless solid, with a molecular formula of C_53_H_82_O_16_, as determined by its ammonium-adducted ion peak ([M + NH_4_]^+^, *m*/*z* = 992.5941, Δ = 0.3 ppm) in the HR-ESI MS spectrum, and the observed carbon signals in the ^13^C NMR spectrum. The ^1^H and ^13^C NMR spectra of **1** exhibited strong similarities to those of dinophysistoxin-1 (DTX-1) isolated from *P. lima*, suggesting that compound **1** is a derivative of DTX-1. Detailed analysis using 1D and 2D NMR spectra (Appendix A) allowed for the assignment of chemical shifts in the DTX-1 framework of compound **1**, which closely matched the shifts observed for DTX-1, except for carbons corresponding to C-1 to C-4 and C-39 (Appendix A). The additional moiety, inferred from the molecular formula to be C_8_H_13_O_3_, was characterized by one methyl (δ_C_ 16.0), two olefinic (δ_C_ 116.1 and 154.0), one methylene (δ_C_ 32.8), three oxymethylene (δ_C_ 59.4, 62.1, and 68.6), and one carbonyl carbon (δ_C_ 167.8), as analyzed based on the ^13^C and HSQC spectra (Table 1). The three deshielded carbon signals (δ_C_ 154.0, 116.1, and 167.8) suggested the existence of an α, β-unsaturated carbonyl functional group, which was supported by the UV absorption band at 214 nm in the IR spectrum. The structure of the additional moiety was elucidated using COSY and HMBC correlations, as illustrated in Figure 2a. The HMBC correlation between H-5′ and C-4′ established the linkage between the α, β-unsaturated carbonyl group and a 1,3-propandiol unit. The ester linkage between C-1′ and the DTX-1 core was confirmed by the HMBC correlation between H-1′ and carbonyl carbon within DTX-1. Lastly, the geometry of the double bond at C-2′ was determined to be in the *E* configuration based on the DP4+ analysis, which predicted 100% probability. The conformational conformers for *E* and *Z* isomers of the diol fragment were optimized by the DFT method at the B3LYP/6-31G(d,p) level, and the NMR shielding tensors for optimized conformers within 4 KJ/mol were calculated using DFT method at the MPW1PW81/6-311G(d,p) with the PCM model in MeOH. DP4+ analysis of the calculated and experimental ^1^H and ^13^C chemical shifts was conducted using the Excel-based program provided by Sarotti (Appendix A) [25].

Compound **2** was identified with the molecular formula C_55_H_84_O_14_, as confirmed by its ammonium-adducted ion peak ([M + NH_4_]^+^ = 986.6167) (Appendix A) and the ^13^C NMR spectrum. The ^1^H NMR spectrum of **2** was closely similar to that of **1**, though it displayed notable differences, particularly at 0.95 ppm and within the deshielded range. Analysis of 1D and 2D NMR spectra (Appendix A) revealed that compound **2** is a derivative of okadaic acid (OA). The ^13^C NMR and HSQC spectra indicated that the structural fragment attached to OA consisted of one methyl (δ_C_ 13.8), one methine (δ_C_ 123.6), two methylene (δ_C_ 34.5 and 41.9), two oxymethylene (δ_C_ 67.5 and 68.7), two exomethylene (δ_C_ 112.9 and 114.5), and three non-protonated carbons (δ_C_ 137.9, 143.0, and 146.7). The connectivity of these carbons was determined using COSY and HMBC correlations, as shown in Figure 2b. The geometry of Δ^6′^ was determined to be in *E* form by the ROE correlation between H-5′ and H_3_-9′. Thus, **2** was named as (*E*)-8′-hydroxy-7′-methyl-2′,4′-dimethyleneoct-6′-enyl okadate, classified as an OA C11 diol.

Compound **3** was characterized with a molecular formula of C_53_H_82_O_14_, supported by its ammonium-adducted ion [M + NH_4_]^+^ = 960.6043 (Appendix A) and the ^13^C NMR spectrum. The ^1^H NMR spectrum of **3** showed a close resemblance to that of **2**, although the ^13^C NMR spectrum revealed a different number of carbons. Compound **3** was identified as a derivative of OA, belonging to the C9-diol class, based on COSY and HMBC correlations (Appendix A) illustrated in Figure 2c. The geometry of the olefinic group in the fragment was assigned to the *E* configuration by comparing the carbon chemical shift in C-8′ with that observed in **1** (Table 2). This assignment was further validated by DP4+ probability calculation, as performed for **1** (Appendix A). Thus, **3** was determined to be (*E*)-7′-hydroxy-2′-methyl-4′-methylenehept-2′-enyl okadate, and was classified as an OA C9 diol.

Compound **4** was determined to have a molecular formula of C_52_H_80_O_14_, as indicated by its ammonium-adducted ion [M + NH_4_]^+^ = 946.5886 (Appendix A) and the ^13^C NMR spectrum. The ^1^H NMR spectra of **3** and **4** were very similar, except for a difference at around 5.9 ppm. Compound **4**, also a derivative of OA, was structurally shorter than **3**, with fewer carbons (Appendix A). Notably, **4** lacked a methyl signal in the C7 chain. The structural features of the fragment were shown in Figure 2d. The geometry of the double bond was assigned as *E* form by the ROE correlation between H-3′ and H-5′, and H-4′ and H-6′. Consequently, **4** was (*E*)-7′-hydroxy-2′-methylenehept-4′-enyl okadate, classified as an OA C8-diol.

### 2.2. Cytotoxicity Assessment of Compounds ***1***–***4***

The cytotoxicity of the diol derivatives (**1**–**4**), isolated from *P. lima*, was tested against three cell lines: HCT116 (human colon cancer cells), Neuro2a (mouse brain neuroblastoma cells), and HepG2 (human liver carcinoma cells) and a comparison was performed, with OA serving as the positive control. Each cell line was exposed to the compounds at concentrations of 0.1, 1, and 10 µM for 24 h (Figure 3). Among the cell lines, Neuro2a showed the highest sensitivity to the compounds. Based on IC_50_ values, compound **3** exhibited the strongest cytotoxicity, with effects similar to those of the positive control, OA (Table 3). The IC_50_ values for **3** were 0.07, 0.17, and 0.17 μM for Neuro2a, HCT116, and HepG2 cells, respectively, while those for OA were 0.07, 0.14, and 0.14 μM. Compounds **2** and **4** exhibited mild cytotoxicity across all three cell lines, with IC50 values ranging from 4.30 to 5.78 μM. Compound **1** showed relatively high cytotoxicity in Neuro2a and HCT116 cells, with IC50 values of 0.10 and 1.54, respectively, but was less toxic to HepG2 cells, where the IC50 exceeded 10 μM. To assess whether the cytotoxicity was due to apoptosis, flow cytometric analysis using Annexin V-FITC/PI staining was performed to detect apoptotic and necrotic cell populations (Appendix A).

## 3. Discussion

Three new OA diol esters and one new DTX-1 diol ester were isolated from the nonpolar cytotoxic fraction of laboratory-cultured *P. lima*. The structures of all compounds were elucidated through detailed analysis of 1D and 2D NMR spectra, supported by MS data. Compound **1**, a DTX-1 derivative, was notable due to the presence of an additional ester group within the diol unit, marking the first discovery of a diester diol derivative of DTX-1. This unique structure represents a new framework distinct from previously reported OA or DTX-1 diol esters. This functional group likely forms through initial oxidation at C-4′ followed by esterification with 1,3-propandiol. Compound **2** is the first OA derivative identified with a C11-diol structure. Previously reported OA and DTX-1 diol or triol derivatives typically feature esterification with C6 to C10 carbon unit and an exomethylene or an olefinic methyl group at C-2′. Some also exhibited additional methyl or exomethylene branch along the carbon chain (Appendix A). In contrast, compound **2** exhibited three branched carbons along a linear carbon chain, diverging from the conventional OA and DTX-1 diol ester structures. New structural characterizations were also determined for compound **3** (OA C9-diol) and compound **4** (OA C8- diol), expanding components of OA diol toxins relevant for toxin monitoring. The discovery of compounds **1** and **2** substantially broadens the spectrum of OA/DTX diol toxins, offering valuable insights into the structural diversity of these toxin derivatives. While LC-MS/MS is widely employed for detecting new OA/DTX-1 derivatives, isolation and structural characterization of these compounds require detailed and comprehensive analyses to refine and expand the existing toxin profiles.

In this study, the cytotoxicity of newly identified diol derivatives was evaluated by determining IC_50_ values and comparing them to OA using widely employed cell lines for cytotoxicity assessment, namely Neuro2a, HCT116, and HepG2. Among the isolated diol derivatives, compound **3** demonstrated the highest cytotoxicity, with levels comparable to OA across all three cell lines. The cytotoxic effects of the four diol derivatives varied among the three cell lines, showing similar or weaker toxicity than OA. Previous research by Wu et al. examined the toxicity in mice with hydrolyzed and unhydrolyzed *P. lima* cells [21]. Unhydrolyzed cells, which contain diol esters, demonstrated higher toxicity, indicating that these diol esters contribute to the overall toxic effects of *P. lima*. Our findings indicate that the toxicity of *Prorocentrum* species may vary among strains, likely reflecting differences in their specific diol ester compositions. To further investigate this relationship, future studies should explore the correlation between the toxicities of different *P. lima* strains and their toxin profiles.

## 4. Materials and Methods

### 4.1. Instrumentation

Optical rotation was measured on a JASCO P-1010 polarimeter (Jasco, Tokyo, Japan) using a 200 µL cell. The UV spectrum was recorded on a Varian Cary 50 spectrophotometer (Varian, Palo Alto, CA, USA), and the IR spectrum was obtained using a JASCO FT/IR 4100 spectrometer (Jasco, Tokyo, Japan). NMR spectra were measured in MeOH-*d*_4_ (with residual solvent peaks at δ_H_ 3.30 and δ_C_ 49.0) using a Bruker Avance II 900 MHz (Bruker BioSpin, Rheinstetten, Germany) at the Korea Basic Science Institute (KBSI) and a Varian VNMRS 500 MHz spectrometer (Varian, Palo Alto, CA, USA). HRESIMS data were collected using a SCIEX X500R instrument (SCIEX, Framingham, MA, USA). HPLC was performed with an Agilent 1200 system (Agilent, Santa Clara, CA, USA) equipped with an RI detector.

### 4.2. Cultures of P. lima

The same *P. lima* strain used in a previous study was utilized for this investigation [7]. To obtain minor components, an additional 1000 L culture was grown using the same cultivation method and harvested in the late-stationary phase of cell growth. This process yielded approximately 350 g of dinoflagellate material, which was then used for the isolation of derivatives of OA/DTX-1.

### 4.3. Extraction and Isolation of OA and DTX-1 Derivatives

The harvested cells were centrifuged, and the resulting pellet was extracted with 100% MeOH at 25 °C for 24 h. After lyophilization, the cells were extracted with 100% MeOH and then the extract (~20 g) was partitioned between H_2_O and CH_2_Cl_2_. The aqueous fraction was further partitioned with H_2_O and butanol, while the organic phase was subjected to partitioning with 85% aqueous MeOH and hexane. The butanol layer and 85% aqueous MeOH layer were then combined and subjected on fractionation using open-column chromatography on a reversed-phase column. A stepwise gradient elution was performed, starting from 50% H_2_O: 50% MeOH (MR1) and gradually increasing to 100% MeOH (MR6) in 10% MeOH increments, resulting in six fractions labeled MR1 to MR6. Compounds **1**–**4** were isolated from the bioactive MR5 fraction. This fraction underwent reversed-phase silica HPLC, producing eight subfractions (rp1–rp8). For this separation, a semipreparative C8 column was utilized at a flow rate of 2 mL/min with UV detection at 210 nm. The mobile phase consisted of H_2_O (A) and acetonitrile (B), with the volume of B increased from 40% to 100% over 45 min. Subsequently, compound **1** (0.7 mg) was purified at a retention time of 28 min, and the compound **3** (2.2 mg) was purified at 32 min from subfraction rp5 using reversed-phase HPLC on a semipreparative C18 column using 85% aqueous MeOH as the eluting solvent, detected by an RI detector. Similarly, compound **2** (0.7 mg) was purified from rp6, and compound **4** (2.6 mg) was purified from rp4. The quantities of the four isolated compounds were lower than those of OA (3.0 mg) and DTX-1 (5.2 mg).

 **1**: colorless solid, [α]^25^_D_ + 85.0 (*c* 0.06, MeOH); UV (MeOH) λ_max_ (log ε) 214 (3.84) nm; IR ν_max_ 3431, 2926, 1724, 1596, and 1080 cm^−1^; ^1^H and ^13^C NMR data, Table 1 and Table 2; HRESIMS *m*/*z* 992.5938 [M + NH_4_]^+^ (calcd for C_53_H_82_O_16_, 992.5941, Δ = 0.3).

 **2**: colorless solid, [α]^25^_D_ + 97.0 (*c* 0.08, MeOH); IR ν_max_ 3411, 2926, 1739, 1591, and 1080 cm^−1^; ^1^H and ^13^C NMR data, Table 1 and Table 2; HRESIMS *m*/*z* 986.6167 [M + NH_4_]^+^ (calcd for C_55_H_84_O_14_, 986.6199, Δ = 2.7).

 **3**: colorless solid, [α]^25^_D_ + 136 (*c* 0.05, MeOH); UV (MeOH) λ_max_ (log ε) 227 (3.56) nm; IR ν_max_ 3397, 2924, 1739, 1591, and 1358 cm^−1^; ^1^H and ^13^C NMR data, Table 1 and Table 2; HRESIMS *m*/*z* 930.6029 [M + NH_4_]^+^ (calcd for C_53_H_82_O_14_, 960.6043, Δ = 1.4).

 **4**: colorless solid, [α]^25^_D_ + 33.8 (*c* 0.4, MeOH); IR ν_max_ 3351, 2923, 1735, 1647, and 1408 cm^−1^; ^1^H and ^13^C NMR data, Table 1 and Table 2; HRESIMS *m*/*z* 946.5878 [M + NH_4_]^+^ (calcd for C_52_H_80_O_14_, 946.5886, Δ = 0.9).

### 4.4. DF4+ Probability Calculation for the Diol Moiety of Compounds ***1*** and ***3***

A conformational search for the two isomers (*E* and *Z*) was conducted using the MMFF module in the Spartan 20 program. For each isomer, hundreds of accessible conformers generated from the search were filtered to those within a 10 kJ/mol energy range, according to a previously reported protocol. The selected conformers were further optimized using DFT at the B3LYP/6-31G(d,p) level in the Gaussian 16 program. The electronic and thermal free energies of each conformer were calculated, and conformers with a 4 kJ/mol energy threshold were chosen. For these low-energy conformers, ^1^H and ^13^C NMR shielding tensors were calculated using the DFT method at the MPW1PW81/6-311G(d,p) level with the PCM model in MeOH. The NMR shielding tensors of each isomer were then averaged with the weights of the low-energy conformers using Boltzmann distribution. Based on these averaged NMR shielding tensors, a DP4+ probability calculation was performed using a previously reported Excel spreadsheet [24].

### 4.5. Cell Cultures

HCT116 cells (human colon cancer cells) and HepG2 cells (human liver carcinoma cells) were obtained from the Korea Cell Line Bank (Seoul, Republic of Korea), while Neuro2a cells (mouse brain neuroblastoma cells) were purchased from ATCC. The cells were maintained in DMEM supplemented with 10% FBS, penicillin (100 IU/mL), and streptomycin (10 mg/mL), at 37 °C in a humidified atmosphere containing 5% CO_2_ and 95% relative humidity.

### 4.6. Cytotoxicity Assessment

The test compounds were dissolved in DMSO (final concentration of 0.1%) and diluted in serum-free culture medium. Prior to the assay, cells were seeded at the following densities in 96-well plates (100 μL per well) and incubated for 24 h: HCT-116 at 1 × 10^5^ cells/mL, Neuro2a cells: 2 × 10^5^ cells/mL, and HepG2 cell: 5 × 10^4^ cells/mL. Cells were then treated with the vehicle control or test compounds at the specified concentrations for 24 h. The inhibitory effect on cell proliferation was evaluated using the CCK-8 assay. After treatment, 10 μL of CCK-8 solution was added to each well, and the cells were incubated for 2 h. Absorbance was measured at 450 nm using a microplate reader.

### 4.7. Flow Cytometry for Apoptosis Analysis

HCT-116, HepG2, and Neuro2a cells were seeded in 6-well plates and treated with the test compounds at the specified concentrations for 24 h. To analyze apoptosis, cells were detached using a plastic cell scraper, harvested, and resuspended with DMEM containing 1% FBS (dilution buffer) at a concentration of 5 × 10^5^ cells/mL. A mixture of 100 μL Annexin V/dead cell reagent and 100 μL of the cell suspension was prepared and incubated in the dark for 20 min at 25 °C. Apoptosis was then quantified using a MUSE cell analyzer (Merck Millipore, Germany).

## Figures and Tables

**Figure 1 toxins-17-00028-f001:**
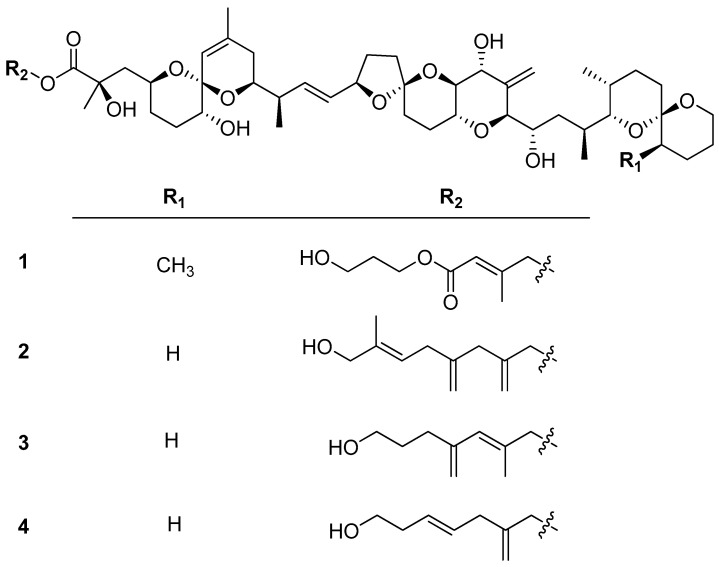
Four new diol derivatives of OA/DTX-1 from the cultures *P. lima*.

**Figure 2 toxins-17-00028-f002:**
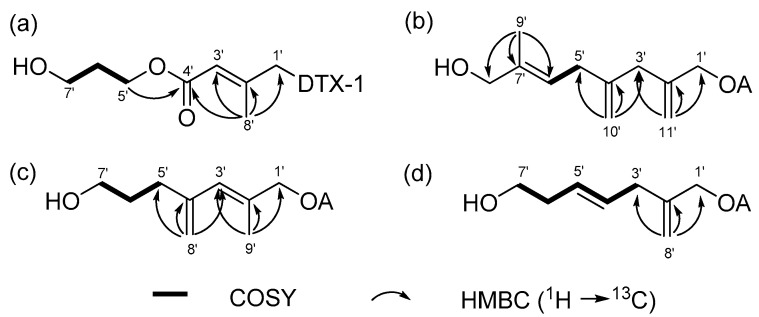
COSY (bold lines), key HMBC (arrows) correlations in the diol moiety of (**a**) compounds **1**, (**b**) **2**, (**c**) **3**, and (**d**) **4**.

**Figure 3 toxins-17-00028-f003:**
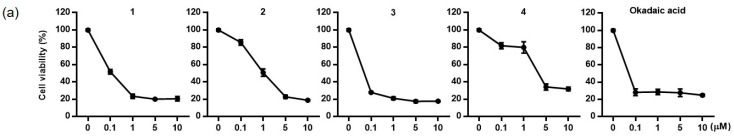
Cytotoxic effects of compounds **1**–**4** and okadaic acid on Neuro-2a (**a**), HCT116 (**b**), and HepG2 (**c**) cells.

**Table 1 toxins-17-00028-t001:** ^13^C NMR data for the diol moiety of compounds **1**–**4** (CD_3_OD).

No	δ_C_, Mult			
	1	2	3	4
1’	68.6, CH_2_	67.5, CH_2_	71.3, CH_2_	68.0, CH_2_
2’	154.0, C	143.0, C	133.8, C	144.5, C
3’	116.1, CH	41.9, CH_2_	130.0, CH	37.7, CH_2_
4’	167.8, C	146.7, C	146.2, C	130.4, C
5’	62.1, CH_2_	34.5, CH_2_	34.6, CH_2_	130.0, C
6’	32.8, CH_2_	123.6, CH	32.3, CH_2_	36.9, CH_2_
7’	59.4, CH_2_	137.9, C	62.4, CH_2_	62.8, CH_2_
8’	16.0, CH_3_	68.7, CH_2_	115.0, C	113.2, CH_2_
9’		13.8, CH_3_	16.0, CH_3_	
10’		112.9, CH_2_		
11’		114.5, CH_2_		

**Table 2 toxins-17-00028-t002:** ^1^H NMR data for the diol moiety compounds **1**‒**4** (CD_3_OD).

No	δ_H_ (Mult, J)			
	1	2	3	4
1’	4.57 (d, 16.9)	4.54 (d, 13.5) 4.64 (d, 13.5) - 2.88, br s - 2.79 (d, 7.3) 5.48 (t, 7.3) - 3.98, s 1.66, s 4.86, s; 4.91, s 5.04, s; 5.17, s	4.54 (d, 12.5)	4.52 (d, 13.4)
	4.70 (d, 16.9)		4.64 (d, 12.5)	4.62 (d, 13.4)
3’	5.90 (q, 1.2)		5.92, br s	2.79 (d, 5.1)
4’	-		-	5.53, m
5’	4.20 (t, 6.4)		2.17 (t, 7.1)	5.53, m
6’	1.87, m		1.62, m	2.25 (dt, 6.2, 6.8)
7’	3.64 (t, 6.1)		3.53 (t, 6.4)	3.56 (t, 6.8)
8’	2.13, s		4.87, br s; 5.03, br s	4.97, br s; 5.05, br s
9’			1.82 (d, 1.2)	
10’				
11’				

**Table 3 toxins-17-00028-t003:** IC_50_ values of compounds **1**–**4** and okadaic acid against Neuro-2a, HCT116 and HepG2 cells.

	μM		
	Neuro2a	HCT116	HepG2
**1**	0.10	1.54	>10
**2**	0.72	5.24	5.78
**3**	0.07	0.17	0.17
**4**	4.54	4.30	5.33
**Okadaic acid**	0.07	0.14	0.14

## Data Availability

The original contributions presented in this study are included in this article and Appendix A. Further inquiries can be directed to the corresponding authors.

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
