# Peer review of "Isolation and Structural Identification of New Diol Esters of Okadaic Acid and Dinophysistoxin-1 from the Cultured Prorocentrum lima"

_toxins, 2025, doi:10.3390/toxins17010028_

Round 1
Reviewer 1 Report
Comments and Suggestions for Authors
English can be improved, examples :
line 6 treats à threats
Line 12 esters à ester
Line 40 region à regions
What is the ratio of these esters compared to OA and DTX1 in the original microalgal culture ?
How does the bulk harvesting procedure affect possible transformations of the diol esters ?
Can it be excluded that these are isolation artefacts (5 isolation steps and little quenching in the bulk harvesting (centrifugation for 10 h, at which temperature ?)
Not answering these questions entertains a doubt as to the biological reality of these compounds and their ecological role.
You cite in your introduction the work by Hu et al., 1995 and 2017 which stipulates that these compounds are self-protective precursors of OA and DTX1. It woul dbe important that you give the concentrations of OA and DTX1 as well your 4 new compounds in the actual algal culture of this P. lima strain and discuss whether you believe that the lack of quenching in the bulk harvesting of cells and storage may explain why these are minor compounds ?
Reviewer 2 Report
Comments and Suggestions for Authors
Please review the attached document.

Some parts are difficult to understand. I recommend reviewing and revising these sections for clarity.
Round 2
Reviewer 1 Report
Comments and Suggestions for Authors
I thank the authors for giving me information in the reply to reviewers concerning the amounts of OA and DTX1 isolated in parallel from this biomass but it is not to me that you to explain this but to the readers of the paper. I didn't find the paper containing this inormation. It should be included (a table would be very interesting comparing the amounts of the different toxins isolated from this biomass) and discussed to improve the intrest of your paper to a wider audience.
Author Response
It should be included (a table would be very interesting comparing the amounts of the different toxins isolated from this biomass) and discussed to improve the interest of your paper to a wider audience.
-> Thank you for your comment. We described the quantities of the four compounds in the Method section of the original version. As you commented, for comparison with OA and DTX-1, we added the quantities of OA and DTX-1 in the revision version.
Reviewer 2 Report
Comments and Suggestions for Authors
Thank you to the authors for their responses. While they have significantly improved the manuscript, there are still some details that need to be addressed.
Introduction:
· Line35- a bracket is missed after OA (DTX-4a, 4b, 5a, 5b, and 5c from OA)
· Although the authors revised the introduction section, the goals and aims of their work remain unclear.
Discussion:
· Although the Journal Guidelines has clearly stated that “Authors should discuss the results and how they can be interpreted in perspective of previous studies and of the working hypotheses.”However, I believe the authors were unable to clearly compare their work with previous studies, as only one study is mentioned in this section.
· Line 183: it should be written as “by Wu et al. [18]…….”

Round 3
Reviewer 1 Report
Comments and Suggestions for Authors
thanks for including the information on the amounts of OA and DTX1 isolated from the same lot of cells